# Identification and Analysis of Immune-Related Gene Signature in Hepatocellular Carcinoma

**DOI:** 10.3390/genes13101834

**Published:** 2022-10-11

**Authors:** Bingbing Shen, Guanqi Zhang, Yunxun Liu, Jianguo Wang, Jianxin Jiang

**Affiliations:** Department of Hepatobiliary Surgery, Renmin Hospital of Wuhan University, 99 Ziyang Road, Wuhan 430060, China

**Keywords:** hepatocellular carcinoma, immune, clinical, mutation, prognosis

## Abstract

Background: Hepatocellular carcinoma (HCC) originates from the hepatocytes and accounts for 90% of liver cancer. The study intends to identify novel prognostic biomarkers for predicting the prognosis of HCC patients based on TCGA and GSE14520 cohorts. Methods: Differential analysis was employed to obtain the DEGs (Differentially Expressed Genes) of the TCGA-LIHC-TPM cohort. The lasso regression analysis was applied to build the prognosis model through using the TCGA cohort as the training group and the GSE14520 cohort as the testing group. Next, based on the prognosis model, we performed the following analyses: the survival analysis, the independent prognosis analysis, the clinical feature analysis, the mutation analysis, the immune cell infiltration analysis, the tumor microenvironment analysis, and the drug sensitivity analysis. Finally, the survival time of HCC patients was predicted by constructing nomograms. Results: Through the lasso regression analysis, we obtained a prognosis model of ten genes including *BIRC5* (baculoviral IAP repeat containing 5), *CDK4* (cyclin-dependent kinase 4), *DCK* (deoxycytidine kinase), *HSPA4* (heat shock protein family A member 4), *HSP90AA1* (heat shock protein 90 α family class A member 1), *PSMD2* (Proteasome 26S Subunit Ubiquitin Receptor, Non-ATPase 2), *IL1RN* (interleukin 1 receptor antagonist), *PGF* (placental growth factor), *SPP1* (secreted phosphoprotein 1), and *STC2* (stanniocalcin 2). First, we found that the risk score is an independent prognosis factor and is related to the clinical features of HCC patients, covering AFP (α-fetoprotein) and stage. Second, we observed that the *p53* mutation was the most obvious mutation between the high-risk and low-risk groups. Third, we also discovered that the risk score is related to some immune cells, covering B cells, T cells, dendritic, macrophages, neutrophils, etc. Fourth, the high-risk group possesses a lower TIDE score, a higher expression of immune checkpoints, and higher ESTIMATE score. Finally, nomograms include the clinical features and risk signatures, displaying the clinical utility of the signature in the survival prediction of HCC patients. Conclusions: Through the comprehensive analysis, we constructed an immune-related prognosis model to predict the survival of HCC patients. In addition to predicting the survival time of HCC patients, this model significantly correlates with the tumor microenvironment. Furthermore, we concluded that these ten immune-related genes (*BIRC5*, *CDK4*, *DCK*, *HSPA4*, *HSP90AA1*, *PSMD2*, *IL1RN*, *PGF*, *SPP1*, and *STC2*) serve as novel targets for antitumor immunity. Therefore, this study plays a significant role in exploring the clinical application of immune-related genes.

## 1. Introduction

Hepatocellular carcinoma (HCC) derives from the hepatocytes and accounts for 90% of liver cancer in general. The global cancer statistics in 2020 show that the mortality of HCC reached 8.3%, ranking the third one in all cancers [1]. The management of HCC generated some changes in recent years. For instance, the major treatment of early HCC is surgical resection, but the primary intrahepatic metastases increase the mortality of HCC patients [2,3,4]. Therefore, it is necessary to explore novel prognostic biomarkers to predict the prognosis of HCC patients.

The immune cell infiltration of the tumor microenvironment affects HCC progression. Generally speaking, aa suppressive tumor microenvironment (TME) restricts antitumor immunity so that HCC patients cannot benefit from immunotherapy [5]. TME is complex and continuously evolving including stromal, fibroblasts, endothelial, and immune cells [6]. At the same time, a previous study showed that tumor immune cell infiltration plays an essential role in tumor development [7]. Additionally, Kole, C et al. emphasized that the tumor microenvironment is vital in response to immunotherapy in HCC patients [8]. Therefore, our study was based on immune-related genes.

In our study, the TCGA and GSE14520 cohorts were utilized to construct the risk model. The immune-related gene signature identified the specific features of immune infiltration, tumor mutation burden, and clinical feature. Although our study was based on the bioinformatic analysis, we claim that the ten immune-related genes (*BIRC5*, *CDK4*, *DCK*, *HSPA4*, *HSP90AA1*, *PSMD*2, *IL1RN*, *PGF*, *SPP1*, and *STC2*) are novel targets for antitumor immunity, which may enrich the field for treating HCC patients.

## 2. Materials and Methods

### 2.1. Data Collection

This study has four databases for collecting data: TCGA (The cancer genome atlas), GEO (Gene Expression Omnibus), IMMPORT (The Immunology Database and Analysis Portal), and TIDE (Tumor Immune Dysfunction and Exclusion). To be more specific, first, the gene expression and clinical files were obtained from TCGA (https://portal.gdc.cancer.gov/, accessed on 19 May 2022) and GEO (https://www.ncbi.nlm.nih.gov/geo/, accessed on 16 August 2021). The TPM (transcripts per million) format was employed in this study in the TCGA database. Furthermore, the GSE14520 cohort was applied [9] to the GEO database. Then, the IMMPORT database (https://www.immport.org/shared/, accessed on 7 September 2022) was applied to obtain 2483 immune genes. Finally, the TIDE score was acquired from the TIDE database (http://tide.dfci.harvard.edu/, accessed on 23 May 2022).

### 2.2. Identification of Differential Immune-Related Genes

The “limma” R package was utilized to conduct the differential analysis with the absolute log2-fold change (|log2FC|) >1 and *p* value < 0.05. The “heatmap” R package was employed to paint the heatmap and volcano graph. R language was applied to acquire the shared genes between the DEGs and 2483 immune-associated genes.

### 2.3. Construction of Immune-Related Genes Prognostics Model

The univariate Cox analysis applies the “survival” package to select the survival-associated genes. Through the lasso regression analysis, the ten-gene prognostics model was constructed. Risk value was calculated by the following equation: Risk score = (expression of *HSPA4* × 0.0406229271055691) + (expression of *HSP90AA1* × 0.179528757102136) + (expression of *PSMD2* × 0.00285728701338196) + (expression of *DCK* × 0.111291761917284) + (expression of *BIRC5* × 0.15170600457355) + (expression of *IL1RN* × −0.00362204612652517) + (expression of *PGF* × 0.0313919529049338) + (expression of *SPP1* × 0.0599218257106274) + (expression of *STC2* × 0.182010671725519) + (expression of *CDK4* × 0.013599196317501). The lasso regression analysis uses the two R packages, covering the “survival” and “glmnet” packages.

### 2.4. Identification and Analysis of Prognosis Model

Survival analysis was conducted by the “survival” and “survminer” packages. Independent prognosis analysis was conducted via the “survival” package. The time–ROC curves were painted by the following packages such as “survival”, “timeROC”, and “survminer”.

### 2.5. Clinical Correlation and Tumor Somatic Mutation Analyses of Prognosis Model

The “ComplexHeatmap” package was employed to construct the clinical association heatmap. The “limma” and “ggpubr” packages were applied to paint the clinical box plot. The Strawberry Perl software was applied to acquire the copy number variations (CNVs) and mutation frequency files. The “maftools” package was utilized to draw the waterfall graph.

### 2.6. Immune Correlation Analysis of Prognosis Model

Single-sample gene set enrichment analysis (ssGSEA) was conducted to identify the differences in the enrichment of 28 immune cells and 13 immune functions, which were acquired from these previous studies [10,11,12]. Additionally, we applied the following calculation methods to evaluate the immune cell infiltration including the TIMER [13], CIBERSORT, CIBERSORT-ABS [14], MCP-counter [15], quanTIseq [16], xCell [17], and EPIC [18] algorithms. The “limma” and “ggpubr” R packages were used to display the differential box plot of the TIDE score. The “reshape2”, “ggplot2”, “limma”, and “ggpubr” R packages were utilized to conduct the immune checkpoint analysis and paint the box plot. In addition, the immune scores were calculated, utilizing the “ESTIMATE” tool by R language [19].

### 2.7. Drug Sensitivity Analysis

We calculated the IC50 values for medications that were collected from the GDSC website (https://www.cancerrxgene.org/, accessed on 9 August 2022). It aims to anticipate prospective compounds employed for HCC treatment. The therapeutic status of medicines was observed by the “oncoPredict” R package in the high-risk and low-risk groups.

### 2.8. Construction and Evaluation of Nomograms

The univariate/multivariate Cox analyses were utilized to assess the prognostic significance of the risk score and clinical features. The R language was employed to paint nomograms and calibration curves in which nomograms were built to predict the 1-, 3-, and 5-year OS, and the calibration curves were pictured to evaluate the facticity of the model.

### 2.9. Statistical Analysis

Statistical analyses were conducted by applying R language (vision 4.2.1), and *p* < 0.05 was deemed as statistically significant.

## 3. Results

### 3.1. Identification of Immune-Related Genes

To begin with, based on the TCGA cohort, we applied R language to conduct the differential analysis and acquired 1557 DEGs. The heatmap (Figure 1A) and the volcano map (Figure 1B) are shown in Figure 1. Furthermore, from the IMMPORT database, 2483 immune-related genes were obtained accordingly. Finally, we received 68 shared genes between 1557 DEGs and 2483 immune-related genes.

### 3.2. Construction of Immune-Related Genes Prognostics Model

Based on the above 68 genes, we selected 16 survival-associated genes through the univariate Cox analysis (Figure 2A). Then, 24 survival-associated genes were identified in the TCGA cohorts. Furthermore, we constructed the ten-genes prognostics model through the lasso regression analysis (Figure 2B, C). In addition, the risk value was calculated by the following equation: Risk score = (expression of *HSPA4* × 0.0406229271055691) + (expression of *HSP90AA1* × 0.179528757102136) + (expression of *PSMD2* × 0.00285728701338196) + (expression of *DCK* × 0.111291761917284) + (expression of *BIRC5* × 0.15170600457355) + (expression of *IL1RN* × −0.00362204612652517) + (expression of *PGF* × 0.0313919529049338) + (expression of *SPP1* × 0.0599218257106274) + (expression of *STC2* × 0.182010671725519) + (expression of *CDK4* × 0.013599196317501).

### 3.3. Identification and Analysis of Prognosis Model

According to the TCGA and GSE14520 cohorts, we painted the time–ROC curve. We unfolded that the AUC values of 1-year, 3-year, and 5-year were 0.777, 0.686, and 0.694, respectively, in the TCGA cohort (Figure 2D); the AUC values were 0.622, 0.661, and 0.686, respectively, in the GSE14520 cohort (Figure 2E). Therefore, the results of our study indicate that the prognosis model possesses a high accuracy, which can be practiced in clinical. Next, we conducted the survival and independent prognosis analyses in the TCGA and GSE14520 cohorts. The results showed that the higher risk score had a worse OS for the HCC patients (Figure 2F,G); the risk score is an independent prognosis factor (Figure 3A–D). Moreover, we painted the heatmap of ten immune-related genes, survival time curve, and survival status graph based on the on TCGA and GSE14520 cohorts (Figure 3E–H), whose results revealed that the higher the risk score, the shorter the survival time.

### 3.4. Clinical Correlation and Tumor Somatic Mutation Analyses of Prognosis Model

First, we analyzed the correlation between the risk score and the clinical features in the TCGA and GSE14520 cohorts. From the heatmap of clinical features (Figure 4A,B), we observed that the risk score was closely related to the stage and level of AFP. From the box plot of clinical features (Figure 4C,D), a statistically significant difference existed in the grade, TNM stage, and level of AFP. The results suggest that the higher the risk score, the higher the malignant level. In addition, we also explored the correlation between gene expression and OS in the TCGA and GSE14520 cohorts. However, we only discovered that the low expression of *LI1RN* was correlated with the worse OS; the high expression of nine genes was associated with worse OS in the HCC patients including *BIRC5*, *CDK4*, *DCK*, *HSP90AA1*, *HSPA4*, *PGF*, *PSMD2*, *SPP1*, and *STC2* (Figure 4E,F).

Subsequently, we focused on the tumor somatic mutation in the prognosis model based on the TCGA cohort. The “maftools” R package was applied to calculate the mutation difference for the low-risk and high-risk groups. From the waterfall graph (Figure 5A,B), we found that a mutation difference existed in the risk groups. The most apparent somatic mutations in *TP53*, *CTNNB1*, and *TTN* existed in the low-risk and high-risk groups. However, we observed that the mutation rate of *TP53* accounted for 33% in the high-risk group and 19% in the low-risk group. Thus, we announced that the somatic mutation of *TP53* was the most obvious out of all of the mutation genes. We also conducted the survival analysis of high-TMB and low-TMB subgroups in the risk model (Figure 5C,D), revealing that high-TMB or high-risk groups had poor OS.

### 3.5. Immune Correlation Analysis of Risk Model

Firstly, we also analyzed the immune functions based on the ssGSEA analysis and used R language to paint the heatmap of immune functions. The result displayed that the difference of immune functions exists in the risk group (Figure 6A). Furthermore, we observed the correlation between the risk score and immune cells based on the seven computational methods: TIMER, CIBERSORT, CIBERSORT-ABS, MCP-counter, quanTIseq, xCell, and EPIC algorithms. The bubble graph (Figure 6B) clearly indicated the correlation between the risk score and immune cells. Subsequently, we concluded the correlation between the risk score and six immune cells based on TIMER (Tumor Immune Estimation Resource). We thought that the risk score is positively related to B cell (Figure 6C), CD4 T cell (Figure 6D), CD8 T cell (Figure 6E), dendritic (Figure 6F), macrophage (Figure 6G), and neutrophil (Figure 6H).

Through our research in this study, we first performed the difference analysis of the immune checkpoint, which revealed that the high-risk group possessed a high expression of immune checkpoints (Figure 6I). Subsequently, TIDE predicts the patient’s response through estimating the published transcriptomic biomarkers based on the tumor pre-treatment expression profiles. We observed that patients in the high-risk group possessed a lower TIDE score, which was more likely to benefit from immunotherapy (Figure 6J–M). Finally, we discovered that the stromal score, immune score, and estimate score were higher in the high-risk than in the low-risk group (Figure 6N–P). Results uncovered that the content of immune and stromal was lower in the low-risk group, and the content of tumor cells was higher in the high-risk group.

Therefore, we concluded that the high-risk group was deemed as the “hot” immune phenotype, with a higher abundance of immune cells and better efficacy for immune checkpoint therapy, but the low-risk group was the “clod” phenotype that featured lower sensitivity to the immunotherapy of the HCC patients.

### 3.6. Drug Sensitivity of Risk Model

The chemotherapeutic response was evaluated in HCC patients utilizing the IC50 values of several chemotherapy drugs. Results revealed that patients in the low-risk group possessed higher sensitivity to multiple drugs including 5-fluorouracil (Figure 7A), VX-11e (Figure 7B), sapitinib (Figure 7C), selumetinib (Figure 7D), sorafenib (Figure 7E), and gemcitabine (Figure 7F), etc. We summarized that 5-fluorouracil, sapitinib, and VX-11e possessed lower IC50 values in the high-risk group. Still, gemcitabine, selumetinib, and sorafenib possessed lower IC50 values in the low-risk group. 5-Fluorouracil, VX-11e, and sapitinib were more effective in the high-risk group; selumetinib, sorafenib, and gemcitabine were more effective in the low-risk group.

### 3.7. Construction and Evaluation of Nomograms

To quantify the individual risk assessment in the HCC patients and better predict the OS of HCC patients, we built nomograms by using four parameters covering the risk score, age, gender, and TNM stage (Figure 7G,H). Based on the scores of each index, the higher total score had a shorter OS. Furthermore, the calibration curve displayed that nomograms are an ideal model and may apply in clinical prediction (Figure 7I,J).

## 4. Discussion

According to the global cancer statistics in 2020, the incidence of HCC ranked seventh, and the mortality of HCC ranked third in all cancers [1]. Therefore, we constructed the immune-related prognosis model to explore novel therapeutic targets and selected HCC patients who gained an advantage from immunotherapy.

First, we obtained 1557 DEGs through differential expression analysis in the TCGA-LIHC-TPM cohort. Based on the IMMPORT database, we acquired 2483 immune-related genes. We summarized and analyzed 68 shared genes between 1557 DEGs and 2483 immune-related genes. Subsequently, we built a prognosis model of ten genes through the lasso regression analysis. The ten genes were *BIRC5*, *CDK4*, *DCK*, *HSP90AA1*, *HSPA4*, *LI1RN*, *PGF*, *PSMD2*, *SPP1*, and *STC2*. In the prognosis model, the TCGA cohort was deemed as the training group, and GSE14520 was regarded as the testing group. The previous research of the ten genes showed:

*BIRC5* was also named surviving, which researchers have paid more attention to as a cancer therapy target [20]. The other study found that *BIRC5* influences the mitosis, apoptosis, and autophagy of the cancer cells [21]. In penile cancer (PC), the silencing *BIRC5* inhibits the inflammatory tumor microenvironment (ITM) and the progression of penile cancer [22]. *CDK4* is a protein of G1/S phase transition in the cell cycle, and CDK4/6 inhibitors have been applied to breast cancer therapy [23]. Meanwhile, R.V. Uzhachenko et al. claimed that CDK4/6 inhibitors delayed the progress of breast cancer and enhanced the recruitment of T cells in a tumor microenvironment [24]. At the same time, the overexpression of *CDK4* is a poor prognostic factor of nasopharyngeal carcinoma (NPC) and influences tumor progression by regulating the p21/CCND1/CDK6/E2F1 signaling pathway [25]. *HSPA4* is a target when B cells selectively drive lymph node metastasis in breast cancer [26]. *HSPA4* is not only involved in colorectal cancer (CRC) progression [27], but also correlates with immune cells in HCC [28]. *DCK* is relevant to drug resistance in cancer [29], but researchers have paid little attention to it. *HSP90AA1* regulated the tumor development by acting as an effective regulator of autophagy in osteosarcoma [30].

*IL1RN* is a competitive antagonist to interleukin-1 (IL1) and involves inflammation regulation. The expression of *IL1RN* is negatively associated with bladder cancer cell proliferation [31]. *PGF* belongs to the pro-angiogenic *VEGF* family, which is correlated with pathological angiogenesis [32]. To be more specific, C.N. Chen et al. observed that *PGF* expression had a relation to the development of gastric cancer [33]; other studies revealed that *PGF* joins in the cancer microenvironment [34,35,36]. *PSMD2* is a member of the *PSMD* gene family [37], which promotes breast cancer cell proliferation through interacting with *p21* and *p27* [38]. *SPP1* connects with tumor-associated macrophage (TAMs) polarization in lung cancer [39]. Furthermore, the silencing *SPP1* inhibits cervical cancer cell proliferation by downregulating the PI3K/Akt signaling pathway [40]. *STC2* is a glycoprotein widely expressed in multiple human tissues and tumor progression [41].

In conclusion, the above studies on the ten immune-related genes (*BIRC5*, *CDK4*, *DCK*, *HSPA4*, *HSP90AA1*, *PSMD2*, *IL1RN*, *PGF*, *SPP1*, and *STC2*) displayed the correlation between the immune-related genes and tumor progression. Therefore, our study summarizes a robust prognostic ability in the ten immune-related genes prognosis model for HCC patients.

According to the survival analysis, independent prognosis analysis, and clinical feature analysis, we deemed that the low-risk scores had a better OS. The risk score is an independent prognostic factor that is closely associated with the clinical features of HCC patients.

TMB (tumor somatic mutation) is deemed as an effective predictor of tumor progression, which features microsatellite instability and represents the mutation frequency of the tumor genome [42]. The higher TMB possessed the higher advantages of immunotherapy in lung cancer treatment [43]. Furthermore, our study found that a difference existed in the TMB of the risk group, and the high TMB and high-risk scores possessed a worse OS. The huge mutation was *TP53* in this study. The *TP53* mutation may represent the response to immunotherapy in lung cancer [44].

TME (tumor immune microenvironment) is involved in tumor progression and affects tumor immunotherapy [45]. The studies revealed that different TME subtypes displayed different advantages in tumor immunotherapy [46,47]. In this study, we carried out immune cell infiltration analysis and then acquired the result that the risk score was relevant to immune cell filtration. Explicitly speaking, we discovered that the risk score was positively related to B cells, CD4 T cells, CD8 T cells, dendritic, macrophages, and neutrophils. Furthermore, we observed that the high-risk group possessed a lower TIDE score than the low-risk group and were more likely to benefit from immunotherapy. We also concluded that the high-risk group had a higher expression of immune checkpoints and possessed higher immune cell enrichment and higher immune activity. Thus, our study may help recognize the unique mechanism of the HCC immune microenvironment, bring about a huge breakthrough in HCC immunotherapy, and develop novel treatment strategies for HCC [48]. Therefore, the high-risk group can be defined as the “hot” immune phenotype, with a higher enrichment of immune cells and better efficacy for immune checkpoint therapy. Nevertheless, the low-risk group is the immune “clod” phenotype, which features lower sensitivity to the immunotherapy of HCC patients [49].

Enhancing the drug sensitivity of HCC treatment could benefit HCC patients [50]. Analyzing the relationship between the risk score and the drug sensitivity, we thought that 5-fluorouracil, sapitinib, and VX-11e possessed lower IC50 values compared to the high-risk group. However, gemcitabine, selumetinib, and sorafenib possessed higher IC50 values in the low-risk group. The lower the IC50 value, the more effective the treatment of drugs [51]. A nomogram integrates the effects of predictors on the outcomes and allows clinicians to predict patient survival by evaluating the whole effect [52]. In our study, nomograms were utilized to quantify the individual risk assessment in HCC patients, which showed the better clinical utility of the signature.

To sum up, our study constructed immune-related gene signatures to predict the prognosis of HCC patients. Nevertheless, there were some limitations to this study. First, all data originated from public datasets, so this study needs other large-scale prospective studies to verify the exploration. In addition, our study was based on bioinformatics analysis and lacked some basic experiments, so further research is urgently needed in the future.

## 5. Conclusions

To conclude, first, through the lasso regression analysis, the prognosis model of the ten immune-related genes was constructed to predict the survival of HCC patients. Furthermore, we announced that the low-risk scores had a better OS through the survival analysis, independent prognosis analysis, and clinical feature analysis. The risk score was an independent prognostic factor, which was closely associated with the clinical features of HCC patients. Finally, through the TMB analysis, immune cell infiltration analysis, and drug sensitivity analysis, the risk score was closely associated with *TP53* mutation, immune cell infiltration, and drug sensitivity. Therefore, we summarize that the ten immune-related genes serve as a novel target for antitumor immunity, which may provide novel insights for the treatment of HCC patients.

## Figures and Tables

**Figure 1 genes-13-01834-f001:**
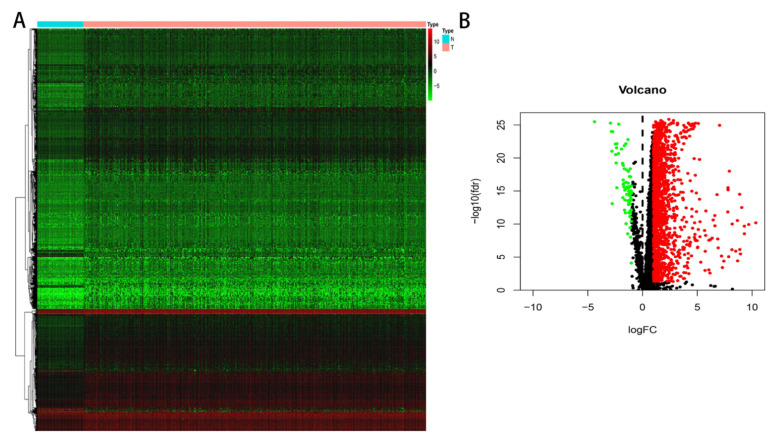
The differential expression analysis in TCGA. (**A**) Heatmap. (**B**) Volcano plot. The green dot represents the low-expression genes, the red represents the high-expression genes and the black represents no differential genes in the volcano.

**Figure 2 genes-13-01834-f002:**
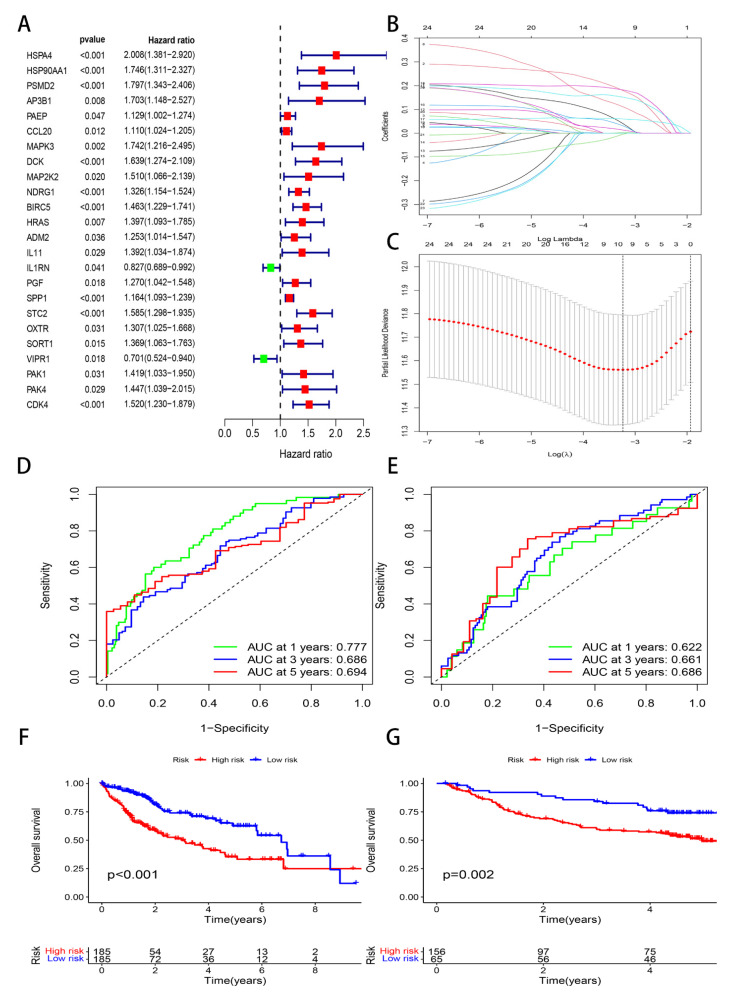
Lasso regression analysis, ROC curve, and Kaplan–Meier survival curves. (**A**) Univariate Cox analysis. (**B**) Distribution of the LASSO coefficients. (**C**) The 10-fold cross-verification of variable selection in the LASSO algorithm. (**D**) ROC curve in the TCGA cohort. (**E**) ROC curve in the GSE14520 cohort. (**F**) Survival curves in the TCGA cohort. (**G**) Survival curves in the GSE14520 cohort.

**Figure 3 genes-13-01834-f003:**
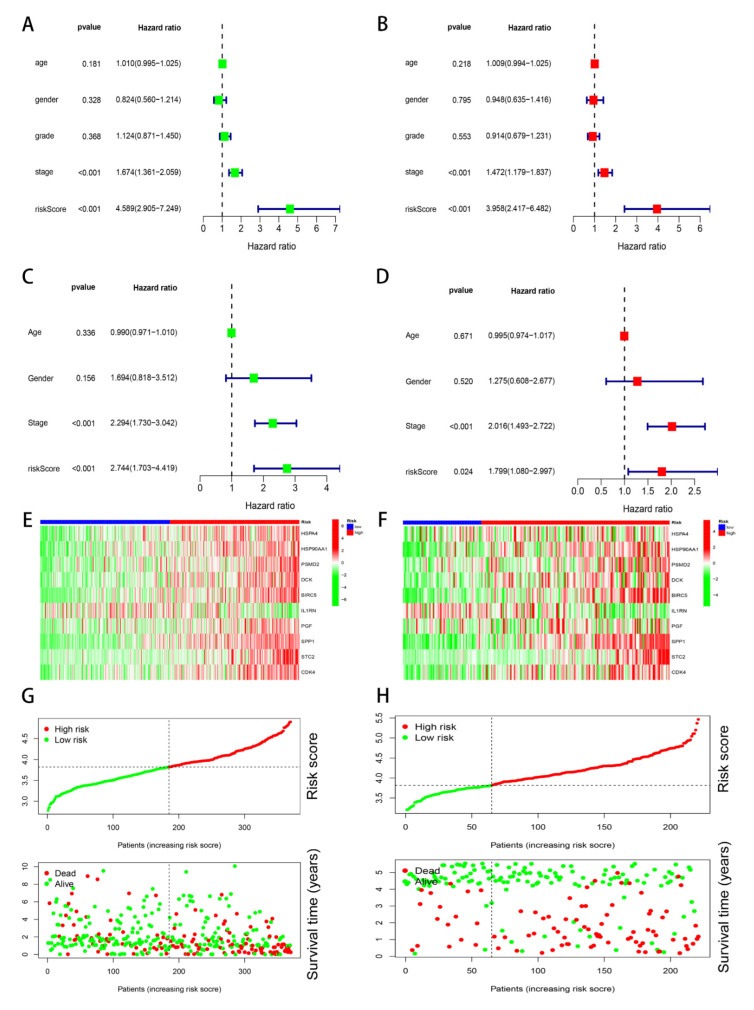
Prognosis of the risk model in two groups. (**A**) Univariate prognosis analysis in the TCGA cohort. (**B**) Multivariate prognosis analysis in the TCGA cohort. (**C**) Univariate prognosis analysis in the GSE14520 cohort. (**D**) Multivariate prognosis analysis in the GSE14520 cohort. (**E**) Heatmap of the ten genes’ expression in the TCGA cohort. (**F**) Heat maps of the ten genes’ expression in the GSE14520 cohort. (**G**) The overall survival risk scores, survival time, and survival status distribution in the TCGA cohort. (**H**) The distribution of the overall survival risk scores, survival time, and survival status in the GSE14520 cohort.

**Figure 4 genes-13-01834-f004:**
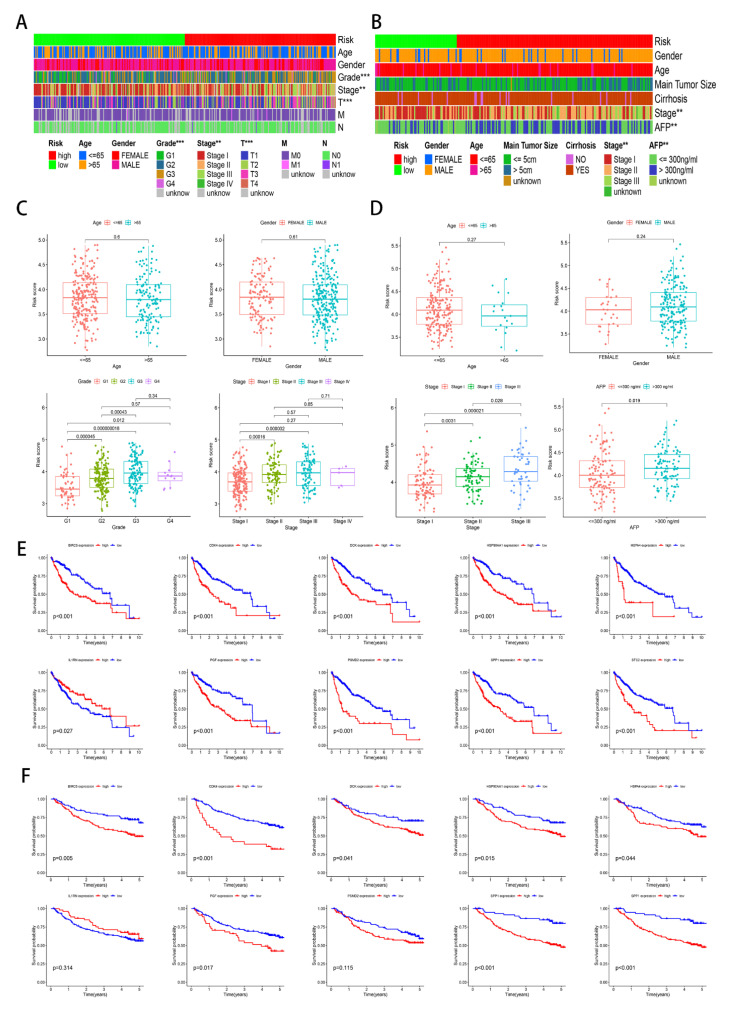
Clinical feature analysis and survival analysis. ** *p* < 0.01; *** *p* < 0.001. (**A**) Heatmap of the clinical correlation in the TCGA cohort. (**B**) Heatmap of the clinical correlation in the GSE14520 cohort. (**C**) Box plot of the clinical correlation in the TCGA cohort. (**D**) Box plot of the clinical correlation in the GSE14520 cohort. (**E**) The OS of the ten genes in the TCGA cohort. (**F**) The OS of ten genes in the GSE14520 cohort.

**Figure 5 genes-13-01834-f005:**
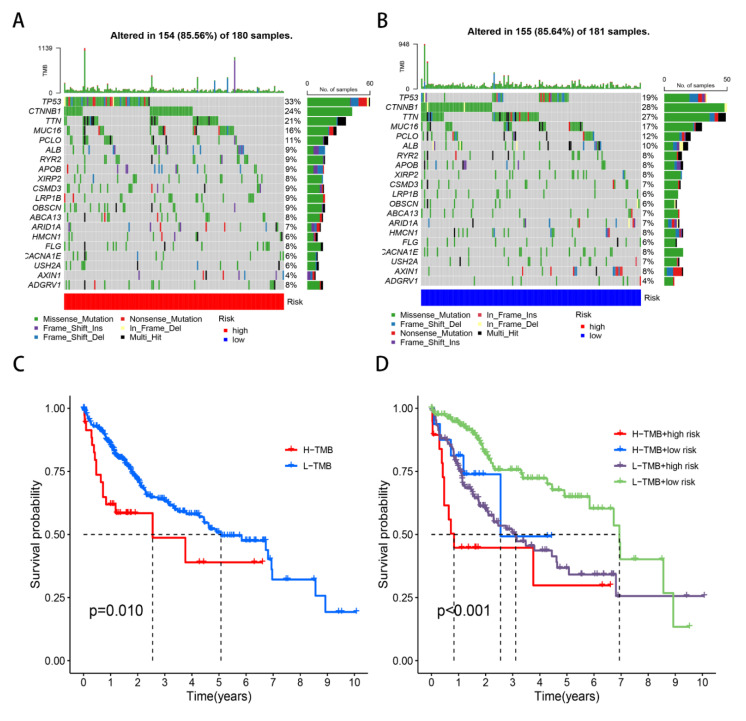
Tumor mutation analysis in TCGA cohort. (**A**) Waterfall plots in the high-risk group. (**B**) Waterfall plots in the low-risk group. (**C**) Survival curves of two groups. (**D**) Survival curves of four groups.

**Figure 6 genes-13-01834-f006:**
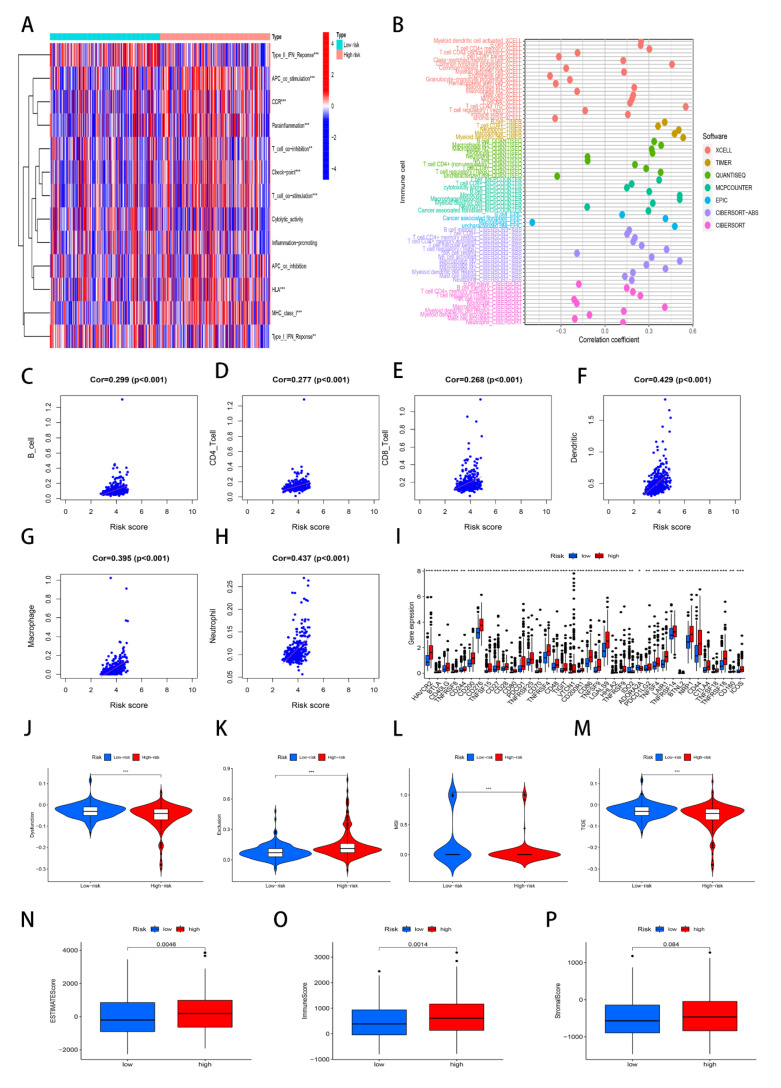
Immune cell infiltration analysis. * *p* < 0.05; ** *p* < 0.01; *** *p* < 0.001. (**A**) ssGSEA scores of immune cells and immune function in the risk group. (**B**) Immune cell bubble plot. (**C**–**H**) Correlation between the risk score and six immune cells. (**I**) The expression of immune checkpoints. (**J**–**M**) Violin graph of TIDE score, MSI, dysfunction, and exclusion scores between the low- and high-risk groups, respectively. (**N**–**P**) Box graphs of ESTIMATEScore, ImmuneScore, and StromalScore between the low- and high-risk groups, respectively.

**Figure 7 genes-13-01834-f007:**
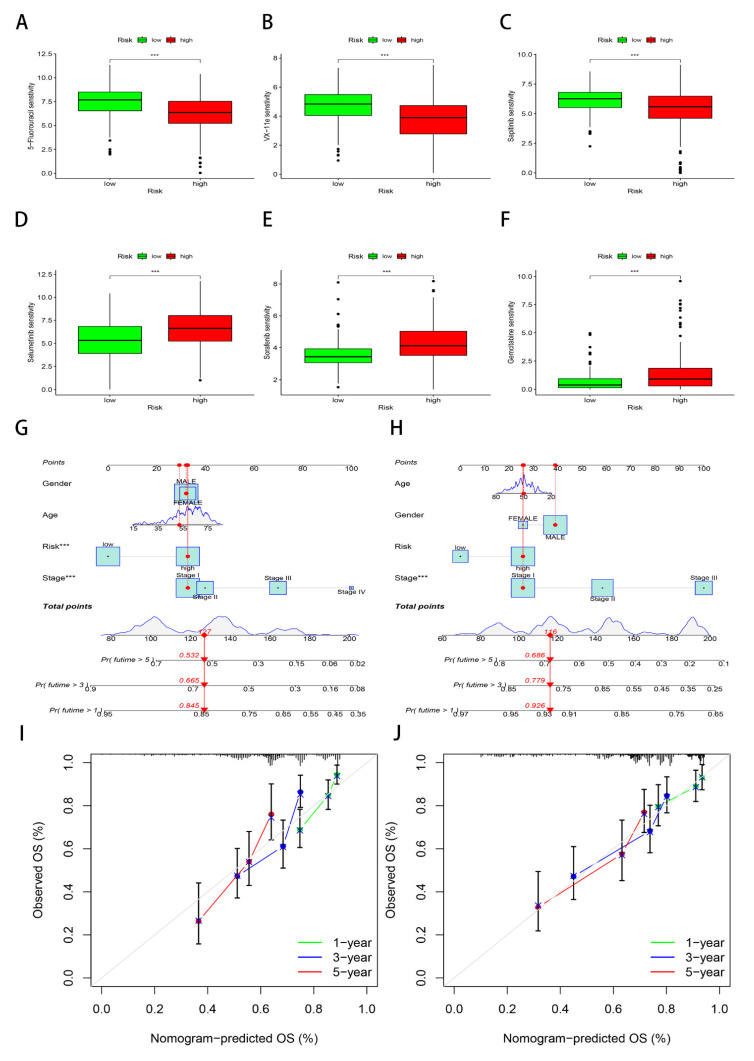
Drug sensitivity analysis and nomogram. *** *p* < 0.001. (**A**–**F**) Drug sensitivity. 5-Fluorouracil (**A**), VX-11e (**B**), and sapitinib (**C**) were more effective in the high-risk group. Selumetinib (**D**), sorafenib (**E**), and gemcitabine (**F**) were more effective in the low-risk group. (**G**) Nomogram in the TCGA cohort. (**H**) Nomogram in the GSE14520 cohort. (**I**) Calibration curves in the TCGA cohort. (**J**) Calibration curves in the GSE14520 cohort.

## Data Availability

The data that support the findings of this study are available in TCGA (https://portal.gdc.cancer.gov/, accessed on 19 May 2022) and GEO (https://www.ncbi.nlm.nih.gov/geo/, accessed on 16 August 2021).

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
