# Peer review of "Identification and Analysis of Immune-Related Gene Signature in Hepatocellular Carcinoma"

_genes, 2022, doi:10.3390/genes13101834_

Round 1
Reviewer 1 Report
Prediction of cancer progression and prognosis is important for the management and treatment of cancer. This study used a regularized model to construct a framework to predict the progression of hepatocellular carcinoma and associated tumor microenvironment changes with tumor prognosis.
The prediction model is the highlight for this study. The authors used careful regularization to prevent overfitting and improve model robustness and the 10 markers are likely to reveal new tumor biology. Can the authors show the prediction results of excluding each marker in the model and compare the robustness of the overall prediction? Are there known drug targets that are correlated with the identified markers and may replace the markers in the model? Answering these questions would better support the significance of this study.
Author Response
Point-to-point Response
We are grateful for your careful and constructive comments on our paper. Your insightful comments were really helpful and led to improved clarity and impact of this work. Also, the text has been clarified and toned down where appropriate.
Attached is the Point-by-Point Response letter.
Reviewer: Prediction of cancer progression and prognosis is important for the management and treatment of cancer. This study used a regularized model to construct a framework to predict the progression of hepatocellular carcinoma and associated tumor microenvironment changes with tumor prognosis. The prediction model is the highlight for this study. The authors used careful regularization to prevent overfitting and improve model robustness and the 10 markers are likely to reveal new tumor biology. Can the authors show the prediction results of excluding each marker in the model and compare the robustness of the overall prediction? Are there known drug targets that are correlated with the identified markers and may replace the markers in the model? Answering these questions would better support the significance of this study.
Response:
Thanks to you for your professional comments.
In our study, based on the immune-associated genes, we constructed a model to predict the progression of hepatocellular carcinoma. Based on the model, we identified the “hot” immune phenotype and immune "clod" phenotype, which displayed better prediction for selecting HCC patients who were beneficial in immunotherapy. In addition, the model also showed better prognosis features to predict the survival rate of HCC patients based on the Nomograms. Additionally, ROC curves and Nomograms evaluated the robustness of the overall prediction.
Furthermore, some studies found that drug targets are correlated with the identified markers. BIRC5 also was known as surviving, which is a therapeutic cancer target. Li, F. et al. (PMID: 31439015) classified surviving cancer treatment into five categories, covering protein interaction inhibitors, homodimerization inhibitors, gene transcription inhibitors, mRNA inhibitors, and surviving immunotherapy. CDK4/6 inhibitors (PMID: 27017286) were allowed to utilize clinical treatment of breast cancer, including palbociclib (PD-0332991), ribociclib (LEE011), and abemaciclib (LY2835219). DCK (deoxycytidine kinase) overexpression enhances the sensitivity of gemcitabine in meningioma cells, which indicates that DCK is a treatment target for tumor therapy (PMID: 33556172). In osteosarcoma chemotherapy, doxorubicin, cisplatin, and methotrexate commonly increased the HSP90AA1 expression, which reduced the sensitivity of osteosarcoma cells to chemotherapy (PMID: 30153855). According to the above reports, we speculated that there are correlations between immune-associated genes and drug targets, which suggests drug targets may replace those markers in this model. However, relevant studies were still lacking and were urgently needed in the future.

Reviewer 2 Report
Sheng et al. studied how to provide new knowledge for HCC treatment. They work is well written and it is sound very interestingly.
However, I would like to suggest some little revision to improve the lecture of their work. The Figure 1A is too little. I suggest to add a supplmentary figure in which is possibile to consult one bigger and with better resolution.
Author Response
Point-to-point Response
We are grateful for your careful and constructive comments on our paper. Your insightful comments were really helpful and led to improved clarity and impact of this work.
Attached is the Point-by-Point Response letter.
Reviewer
Sheng et al. studied how to provide new knowledge for HCC treatment. They work is well written and it is sound very interestingly.
However, I would like to suggest some little revision to improve the lecture of their work. The Figure 1A is too little. I suggest to add a supplmentary figure in which is possibile to consult one bigger and with better resolution.
Response
We are very sorry for our negligence of the problem. According to your suggestion, we have replaced Figure 1A with higher magnification and clearer images. If I have made some mistakes about my modification, I will correct them in time. Thank you very much.

Reviewer 3 Report
A number of issues to be addressed by the authors:
- The aim of the study is really poorly presented in the abstract ("Our study aims to provide novel insight for HCC treatment" is too vague and completely non-specific) and not at all presented in the end of the introduction.
- The abstract should be improved "The novel nomogram prediction model has high accuracy" again is really general and does not provide any statistical information for the readership and the content of this study.
- Do not analyze steps taken in your introduction section its completely irrelevant.
- Some sentences in the discussion just repeat the outcomes and do not discuss them in the context of the published literature "Finally, a nomogram was utilized to quantify the individual risk assessment in HCC patients, and the novel nomogram prediction model has high accuracy." Again vague and general sentences.
- A limitations section is indespensable and needs to be added.
Author Response
Point-to-point Response
We are grateful to you for your careful and constructive comments on our paper. Your insightful comments were really helpful and led to improved clarity and impact of this work.
Attached is the Point-by-Point Response letter.
Reviewer:
A number of issues to be addressed by the authors:
- The aim of the study is really poorly presented in the abstract ("Our study aims to provide novel insight for HCC treatment" is too vague and completely non-specific) and not at all presented in the end of the introduction.
Response:
We are very sorry for improper writing about this sentence ("Our study aims to provide novel insight for HCC treatment"). Based on your comments, we have replaced the sentence with a more appropriate description in lines 10-11. (Therefore, the identification of novel biomarkers in HCC patients is essential for predicting the prognosis of HCC.)
- The abstract should be improved "The novel nomogram prediction model has high accuracy" again is really general and does not provide any statistical information for the readership and the content of this study.
Response:
Thank you for your professional suggestion. According to your suggestion, we redescribed this sentence in lines 25-26 (Nomogram, including clinical features and risk signature, displayed the better clinical utility of the signature.). If I have made some mistakes about my modification, I will correct them in time. Thank you very much.
- Do not analyze steps taken in your introduction section its completely irrelevant.
Response:
We thank you for the positive comments regarding the manuscript. In response, we removed them and rewritten the introduction section in lines 36-47.
- Some sentences in the discussion just repeat the outcomes and do not discuss them in the context of the published literature "Finally, a nomogram was utilized to quantify the individual risk assessment in HCC patients, and the novel nomogram prediction model has high accuracy." Again vague and general sentences.
Response:
Thank you for pointing this out and constructive suggestion. We added relevant references and rewritten the discussion section. (Nomogram integrates the effects of predictors on outcomes and allows clinicians to predict patient survival by evaluating the sum of the effects. In our study, Nomograms were utilized to quantify the individual risk assessment in HCC patients, which showed the better clinical utility of the signature.)
- A limitations section is indespensable and needs to be added.
Response:
We thank the reviewer for raising this issue. According to your suggestion, we have added the limitations of our study in the discussion section in lines 314-318. If I have made some mistakes about my modification, I will correct them in time. Thank you very much.

Round 2
Reviewer 3 Report
While the authors have addressed the majority of my comments, the language throughout the manuscript needs to be improved.
For instance in the very first sentences of the abstract:
" The incidence and mortality of HCC are relatively high in all cancers" this sentence doesn't make sense, equally the following: "The tumor microenvironment involves in tumor progression" doesn't make any sense either.
Another example "Although HCC patients possess(?) multiple treatment methods, the prognosis is still poor, especially for advanced HCC patients. One essential reason is response to immunotherapy". The first sentence makes no sense; patients with HCC have different treatment options based on patient, tumor characteristics and underlying liver function to be more precise. Also patients with HCC have poor survival because of advanced stage at presentation AND resistance to MULTIPLE regimens as it is a largely chemo- and immuno-resistant tumor.
I urge the authors to critically revise their work, which although may be well conducted yet remains still poorly presented.
Author Response
Point-to-point Response
We are grateful for your careful and constructive comments on our paper. Your insightful comments were helpful and improved the clarity and impact of this work.
Attached is the Point-by-Point Response letter. Reviewer text in red, our answers in black.
Reviewer:
While the authors have addressed the majority of my comments, the language throughout the manuscript needs to be improved.
For instance in the very first sentences of the abstract:
" The incidence and mortality of HCC are relatively high in all cancers" this sentence doesn't make sense, equally the following: "The tumor microenvironment involves in tumor progression" doesn't make any sense either.
Another example "Although HCC patients possess(?) multiple treatment methods, the prognosis is still poor, especially for advanced HCC patients. One essential reason is response to immunotherapy". The first sentence makes no sense; patients with HCC have different treatment options based on patient, tumor characteristics and underlying liver function to be more precise. Also patients with HCC have poor survival because of advanced stage at presentation AND resistance to MULTIPLE regimens as it is a largely chemo- and immuno-resistant tumor.
I urge the authors to critically revise their work, which although may be well conducted yet remains still poorly presented.
Response:
We are very sorry for our negligence of the problem. According to your suggestions, we conducted point-to-point response and revised each one.
Firstly, we polished and improved the entire manuscript through a colleague fluent in English writing. Secondly, for the examples you pointed out, we performed positively modified, and rewritten them. The modifications were shown in lines 9-10 and 37-47. Lastly, we also corrected the other manuscript contents that might be deemed as same as the above modifications.
Finally, we are sorry again for our negligence. Of course, if we make other modification mistakes, we will correct them in time. Thank you very much.
